# Magnetocaloric Effect, Magnetoresistance of Sc$_{0.28}$Ti$_{0.72}$Fe$_2$, and Phase Diagrams of Sc$_{0.28}$Ti$_{0.72}$Fe$_{2-x}$T$_x$ Alloys with $T$ = Mn or Co

**Liting Sun, Hargen Yibole, Ojiyed Tegus and Francois Guillou *** 

Inner Mongolia Key Laboratory for Physics and Chemistry of Functional Materials,
Inner Mongolia Normal University, Hohhot 010022, China; sunlightzd@gmail.com (L.S.);
hyibole@imnu.edu.cn (H.Y.); tegusph@imnu.edu.cn (O.T.)
* Correspondence: francoisguillou@imnu.edu.cn

**Abstract:** (Sc,Ti)Fe$_2$ Laves phases present a relatively unique case of first-order ferro-ferromagnetic transition originating from an instability of the Fe moment. In addition to large magnetoelastic effects making them potential negative thermal expansion materials, here, we show that Sc$_{0.28}$Ti$_{0.72}$Fe$_2$ and related alloys also present sizable magnetocaloric and magnetoresistance effects. Both effects are found substantially larger at the ferro-ferromagnetic transition ($T_{t1}$) than near the Curie temperature $T_C$, yet they remain limited in comparison to other classes of giant magnetocaloric materials. We suggest a strategy to improve these properties by bringing the transition at $T_{t1}$ close to $T_C$, and test its possible realization by Co or Mn for Fe substitutions. The structural and magnetic phase diagrams of Sc$_{0.28}$Ti$_{0.72}$Fe$_{2-x}$T$_x$ alloys with $T$ = Mn or Co are explored. Substitutions for Fe by adjacent Mn or Co elements give rise to a breakdown of the long-range ferromagnetic order, as well as a swift disappearance of finite moment magnetism.

**Keywords:** magnetocaloric effect; magnetoresistance; Laves phases; phase diagrams

---

## 1. Introduction

Laves phases, $AB_2$, form a particularly large materials family exhibiting a vast array of intriguing properties, including those related to magnetism [1,2]. Transition-metal-based Laves phases $AT_2$ (with $A$ an early 3, 4, or 5$d$ transition metal and $T$ a late 3$d$ transition metal) are a typical example of this richness in properties since the crystal, electronic, and magnetic structures of binary and ternary alloys are highly sensitive to chemical composition. Hexagonal C14 transition-metal-based Laves phases such as Hf$_{1-x}$Ta$_x$Fe$_2$ or Hf$_{1-x}$Nb$_x$Fe$_2$ show a first-order ferromagnetic (FM) to antiferromagnetic (AFM) transition when increasing the temperature and have attracted particular interest in recent years for the specific mechanisms of this transition [3–11] and the associated magnetocaloric [4,12], giant negative expansion [9], or magnetoresistance effects [5]. On the other hand, Sc$_{1-x}$Ti$_x$Fe$_2$ Laves phases were intensively studied from the 1970s to 1990s for their complex magnetic phase diagram, in particular the compositions with $0.6 \leq x \leq 0.75$ showing a ferromagnetic–paramagnetic transition around 300–400 K, and an additional, and quite unique, discontinuous ferromagnetic–ferromagnetic transition ($T_{t1}$) around 50–120 K with a change in amplitude of the magnetic moment [13–16]. This FM–FM transition is associated with a large magnetoelastic effect and corresponds to an increase upon cooling of the Fe magnetic moments from ~0.9 $\mu_B$/Fe to ~1.3 $\mu_B$/Fe, breaking down a local moment description. Local probes such as Mössbauer spectroscopy have shown that the Fe moment in the 2$a$ position is highly dependent on the composition [14], and neutron diffraction experiments revealed that an instability of the Fe moment in the 2$a$ position occurs as a function of temperature [17]. Interestingly, similar Fe

moment instabilities are believed to be at the very heart of the exceptionally strong magnetoelastic first-order transitions and giant magnetocaloric effects observed in MnFe(P,Si) and La(Fe,Si)$_{13}$ materials systems [18,19].

In Sc$_{0.35}$Ti$_{0.65}$Fe$_{1.95}$, the low-temperature FM–FM first-order transition ($T_{t1}$) is accompanied by a large cell-volume contraction upon heating ($\Delta V/V \sim -1.1\%$) [15] and sizable transition entropy $\Delta S_{tr}$ estimated between 8 and 16 Jkg$^{-1}$K$^{-1}$ [20], which makes these materials interesting candidates for negative thermal expansion or magnetocaloric applications. It is, however, needed to find control parameters to adjust $T_{t1}$ and $T_C$ transition temperatures. Ideally, one should aim to make both transitions coincide in order to create a synergy between the magnetic moment fluctuation at $T_{t1}$ and the change in magnetic order at $T_C$, which could potentially lead to a first-order magnetic transition with a large discontinuity in magnetization. Due to the large cell-volume contraction upon heating at $T_{t1}$, it could have been anticipated that negative chemical pressure such as that induced by Mn for Fe substitution would increase the $T_{t1}$ transition temperature. However, while the evolution of $T_{t1}$ and $T_C$ due to alloying in Sc$_{1-z}$Ti$_z$Fe$_2$ ternaries [17,20], high pressure [21], or high magnetic fields [22] have been established, the effect of $3d$ electron count on the Fe sites remains unclear. In particular, $AT_2$ Laves phases are known to present complex structural or magnetic phase diagrams with competitions between different structural phases (C14/C15), different magnetic orders (FM, AFM, paramagnetic, or short-range magnetism, etc.), and different degrees of localization of the magnetic moment. Accurate predictions of the outcome of substitutions are thus virtually impossible at present. In this work, we explore the influence of Mn or Co for Fe substitutions in Sc$_{0.28}$Ti$_{0.72}$Fe$_{2-x}$T$_x$ and establish structural and magnetic phase diagrams of these quaternary alloys.

## 2. Materials and Methods

Sc$_{0.28}$Ti$_{0.72}$Fe$_{2-x}$T$_x$ with $T$ = Mn and Co were prepared by arc-melting in a purified Ar atmosphere of commercial staring materials with purity greater than 99.9% (from the Baotou Research Institute of Rare Earths (Baotou, China) for Sc, and from Alfa Aesar (Haverhill, MA, USA) for the other transition metals). Each button is melted and stirred 5 times. For the parent composition Sc$_{0.28}$Ti$_{0.72}$Fe$_2$, the structural and magnetic properties were compared for samples: (*i*) as-cast, (*ii*) annealed at 1000 °C for 4 days and furnace cooled, and (*iii*) annealed at 1000 °C for 4 days and quenched. The three heat treatments led to comparable cell parameters, magnetic saturation, and transition temperatures. The first-order FM–FM transition $T_{t1}$ is, however, broader for (*ii*) furnace cooling than for (*i*) and (*iii*) as-cast and quenched samples, the latter two being on par with each other. Accordingly, for most compositions, only as-cast samples are reported, with a few exceptions indicated in the manuscript.

Powder X-ray diffraction experiments were carried out in reflection on an Empyrean diffractometer (Panalytical, Almelo, The Netherlands) using Cu K$\alpha$ radiation. A TTK600 (Anton Paar, Graz, Austria) low-temperature chamber was used for temperature-dependent measurements. The powders used for XRD experiments were sieved below 36 µm in diameter. The diffraction patterns were analyzed using Fullprof software (version 7.20) [23], and VESTA software (version 3.4.6) was used for crystal structure representation [24]. Physical properties measurements were carried out using a Verslab system (Quantum Design, San Diego, CA, USA) equipped with a vibrating sample magnetometer or AC resistance options (with 4 points contacted with Cerasolzer® (Kuroda Techno Co. Ltd., Yokohama, Japan) and ultrasonic iron), for magnetization and resistivity measurements, respectively.

## 3. Results and Discussion

### 3.1. Magnetocaloric and Magnetoresitance of Sc$_{0.28}$Ti$_{0.72}$Fe$_2$ Ternary Alloy

Sc$_{0.28}$Ti$_{0.72}$Fe$_2$ was chosen as the starting composition for this study as, according to former reports [17,20], it offers a compromise between relatively high $T_{t1}$-transition temperature and sharpness of the transition. As illustrated in the next subsections, the cell parameters and basic magnetic properties of our Sc$_{0.28}$Ti$_{0.72}$Fe$_2$ sample are in line with former reports [17,20]. Figure 1 shows detailed

magnetic data for $Sc_{0.28}Ti_{0.72}Fe_2$ around $T_{t1}$ and $T_C$. Though not particularly sharp, the $T_{t1}$ transition is a first-order magnetic transition (FOMT), as demonstrated by the relatively large thermal hysteresis of ~10 K. The shift of the transition due to the field $dT_{t1}/dB$ is about +7.1(1) K/T, which is larger than the +4 K/T reported for $Sc_{0.25}Ti_{0.75}Fe_2$ [22] or +3.8 K/T $Sc_{0.35}Ti_{0.65}Fe_{1.95}$ [15]. Within Clausius–Clapeyron formalism, this results in a limited transition entropy $\Delta S_{tr} = -\Delta M/(dT_{t1}/dB)$ ~2.8 Jkg$^{-1}$K$^{-1}$. Given that larger transition entropies were reported [22], this suggests that the strength of the FM–FM transition at $T_{t1}$ is highly sensitive to chemical composition, both the Sc/Ti ratio and Fe sub-stoichiometry. Limited transition entropy also results in a limited magnetocaloric effect as calculated from $M_B(T)$ curves using the Maxwell relation [25], with $\Delta S = -1.4$ and $-2.2$ Jkg$^{-1}$K$^{-1}$ at $T_{t1}$ for 1 and 2 T, respectively. The magnetocaloric effect is conventional, with a negative isothermal entropy change for a positive field change, both at $T_{t1}$ and $T_C$. Around $T_C \approx 325$ K, the shape of the $M_T(B)$ curves is typical of a second-order transition, and it corresponds to a magnetocaloric effect of $\Delta S = -0.4$ and $-0.8$ Jkg$^{-1}$K$^{-1}$ for 1 and 2 T, respectively, spreading over a large temperature range. In comparison to other giant magnetocaloric materials [25], the magnetocaloric effect occurring at $T_{t1}$ and $T_C$ is sizable, yet not among the best candidates known in each temperature range. For instance, in 1 T, a giant magnetocaloric effect of $-10$ up to $-20$ Jkg$^{-1}$K$^{-1}$ can be observed in MnFe(P,Si) and related materials [19,26,27], $-18$ Jkg$^{-1}$K$^{-1}$ in LaFe$_{11.44}$Si$_{1.56}$ [28], $-13$ Jkg$^{-1}$K$^{-1}$ in (Mn,Fe)(Fe,Ni)(Si,Ge,Al) alloys [29–32], or an inverse effect of $+12$ Jkg$^{-1}$K$^{-1}$ in FeRh [33] or $+15$ Jkg$^{-1}$K$^{-1}$ in Ni-Co-Mn-Ti alloys [34–37]. Part of the reason is that the magnetocaloric $\Delta S$ scales as $dM/dT$. At $T_{t1}$, the transition remains relatively sharp, about 15 K in width, but the change in magnetization is limited, approximately 0.3 $\mu_B$/Fe. At $T_C$, the decrease in magnetization upon heating is larger (~0.8 $\mu_B$/Fe) but spread over a much broader temperature range (nearly 150 K), leading to limited $dM/dT$. We thus believe that better magnetocaloric properties could be achieved in derivatives from (Sc,Ti)Fe$_2$ by bringing $T_{t1}$ and $T_C$ close to each other, ideally by achieving a coupling between both transitions.

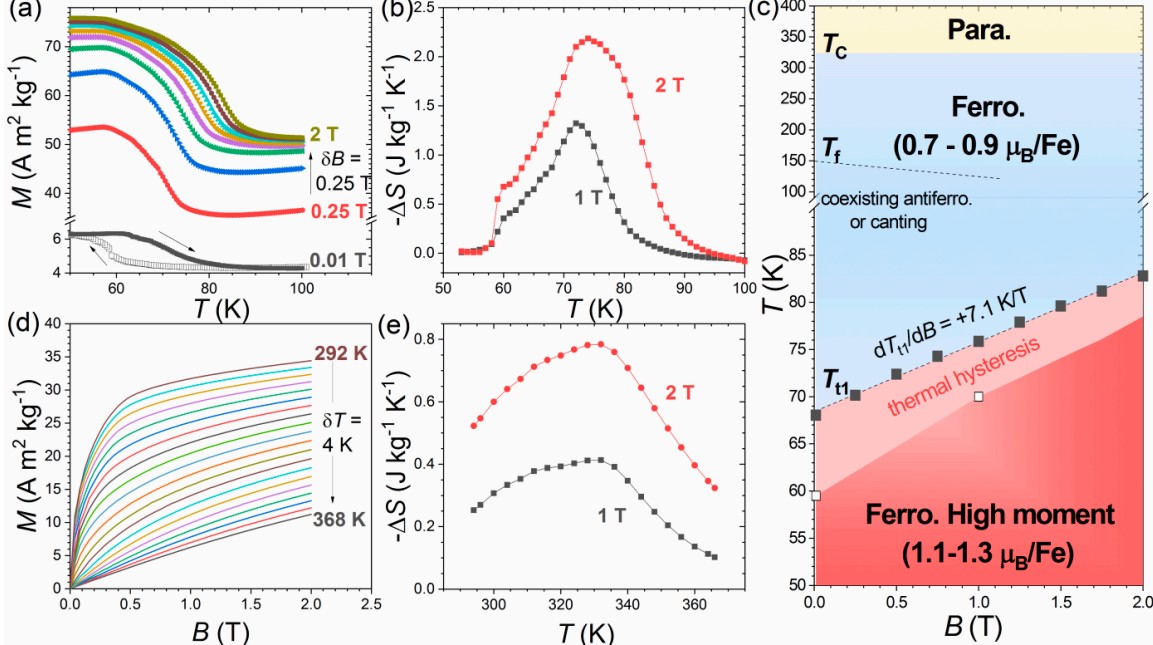

**Figure 1.** Temperature/field dependence of the magnetization and magnetocaloric effect of $Sc_{0.28}Ti_{0.72}Fe_2$. (**a**) $M_B(T)$ curves measured upon heating (full symbols) and cooling (open symbol) around the first-order ferro-ferromagnetic transition $T_{t1}$; (**b**) opposite of the isothermal entropy change near $T_{t1}$; (**c**) magnetic phase diagram; (**d**) magnetization $M_T(B)$ curves at different temperatures near $T_C$; (**e**) opposite of the isothermal entropy change near $T_C$.

To illustrate the very different nature of the magnetic transitions at $T_{t1}$ and $T_C$, electrical resistivity ($\rho$) and magnetoresistance (MR) were measured and are presented in Figure 2. The temperature dependence of the resistivity is metallic, but a linear-like regime is not yet reached on the present data, suggesting a Debye temperature higher than 300 K. $T_C$ does not correspond to any significant $\rho$ anomaly; by contrast, a relatively discontinuous $\rho$ increase of ~30% is observed at $T_{t1}$ with a thermal hysteresis in line with the first-order character of this transition. Due to the magnetic field sensitivity of $T_{t1}$, it results in a significant magnetoresistance whose maximum of −17% for $\Delta B = 3$ T is reached near $T_{t1}$. In this low temperature range, the MR field dependence is also typical of that of a FOMT with a clear field-induced character and a finite magnetic hysteresis. By contrast, the magnetoresistance near $T_C$ is limited, reaching a maximum of only −0.9%, as is usually found in metallic ferromagnets. The reduction in net magnetization at $T_{t1}$ is smaller than that at $T_C$, yet the MR is very significantly larger. This suggests that not only an evolution of magnon–electron interactions takes place at $T_{t1}$, but also that other contributions are involved, such as a change in magnon–phonon terms due to the large magnetoelastic effect occurring at $T_{t1}$.

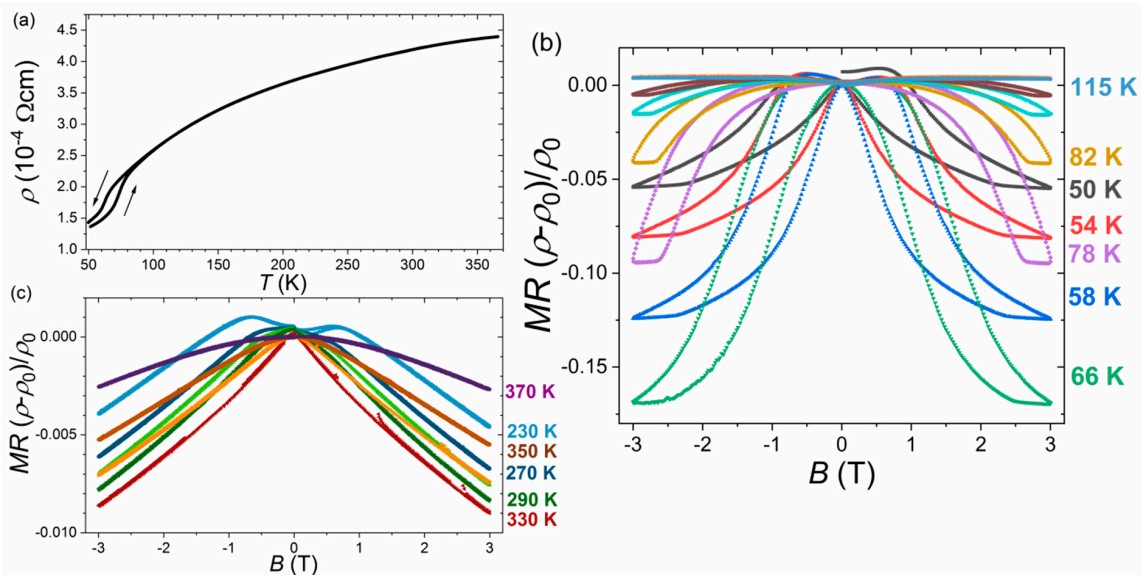

**Figure 2.** (**a**) Temperature dependence of the resistivity for $Sc_{0.28}Ti_{0.72}Fe_2$ in $B = 2$ T. Magnetoresistance at selected temperatures around $T_{t1}$ (**b**) and $T_C$ (**c**).

### 3.2. Co for Fe Substitution in $Sc_{0.28}Ti_{0.72}Fe_2$

Figure 3 presents the powder diffraction patterns of $Sc_{0.28}Ti_{0.72}Fe_{2-x}Co_x$ alloys and the corresponding cell volume and parameters. From $x = 0$ until $x = 1$, all XRD peaks can be indexed in the C14 hexagonal structure. The change from hexagonal to cubic C15 structure occurs around $x = 1.5$. This latter composition shows a coexistence of both phases (~35% hexagonal phase). After 3 days annealing at 1000 °C and quenching to avoid contamination from unreacted CoTi secondary phase, the fully substituted $Sc_{0.28}Ti_{0.72}Co_2$ sample is found to fully crystallize into the cubic C15 structure. Co substitution in $Sc_{0.28}Ti_{0.72}Fe_{2-x}Co_x$ corresponds to a significant cell-volume contraction, in line with the smaller atomic radius of Co than that of Fe. The boundary between hexagonal and cubic structures marks out a large difference in volume (~1.2%). Within the hexagonal phase, the unit cell contraction appears relatively isotropic, as from $x = 0$ to 1 both $a$ and $c$ axes decrease, so that the lattice parameters ratio $c/a$ experiences only a limited increase of about +0.07%.

Figure 4 presents the magnetic properties of $Sc_{0.28}Ti_{0.72}Fe_{2-x}Co_x$ alloys. The unsubstituted sample $x = 0$ shows a Curie temperature near 325 K and the ferro-ferromagnetic transition $T_{t1}$ at 75 K in $B = 1$ T, in line with former reports on stoichiometric $(Sc,Ti)Fe_2$ or Fe-deficient $(Sc,Ti)Fe_{1.95}$ [13–16,20]. In addition, below ~150 K, one can notice a weak decrease in magnetization upon cooling before

reaching $T_{t1}$. This leads to the appearance of a broad bump centered at $T_f$~150 K, also present in former studies, and which was ascribed to the development of a coexisting antiferromagnetic phase or of a canting [17,22]. In the range $0 < x \le 0.5$, Co substitution leads to a rapid decrease in Curie temperature and a strong reduction in magnetization. Alloys with $x > 0.5$ no longer show signs of long-range ferromagnetic order on $M(B)$ or $M(T)$ curves, and no spontaneous magnetization could be observed on Arrott plots at 50 K (not shown). The change in crystal structure does not appear directly responsible for the loss of long-range magnetic order, as finite magnetic moments and ferromagnetism disappear within the compositional range of the C14 crystal structure. The fully substituted alloy $Sc_{0.28}Ti_{0.72}Co_2$ presents a very small magnetization (magnetization at room temperature is four orders of magnitude smaller than paramagnetic $x = 0.5$) and has only limited temperature dependence, recalling the Pauli paramagnetism of its $TiCo_2$ or $ScCo_2$ C15 parents [38,39]. The intermediate compositions with Co partially substituting Fe cannot be compared with closely related C14 ternaries such as $TiFe_{2-x}Co_x$, as Sc plays a critical role in stabilizing ferromagnetism in $Sc_{0.28}Ti_{0.72}Fe_{2-x}Co_x$ with $x \le 0.5$. In addition, in contrast to $TiFe_{2-x}Co_x$ with $x \le 0.6$ [40], no clear signature of long-range antiferromagnetic order is observed. The saturation magnetization at $T = 50$ K and $B = 3$ T is about 1.1 $\mu_B$/Fe, 0.55, 0.43, and 0.32 $\mu_B$ per Fe or Co atom for $x = 0$, 0.1, 0.2, and 0.3, respectively, and the induced magnetization ~$3 \times 10^{-3}$ $\mu_B$/Co for $x = 2$. The decrease in magnetization is faster than what could have been anticipated due to magnetic dilution of Fe by non-magnetic Co. In addition, the FM–FM transition at $T_{t1}$ is no longer observed in Co-substituted samples.

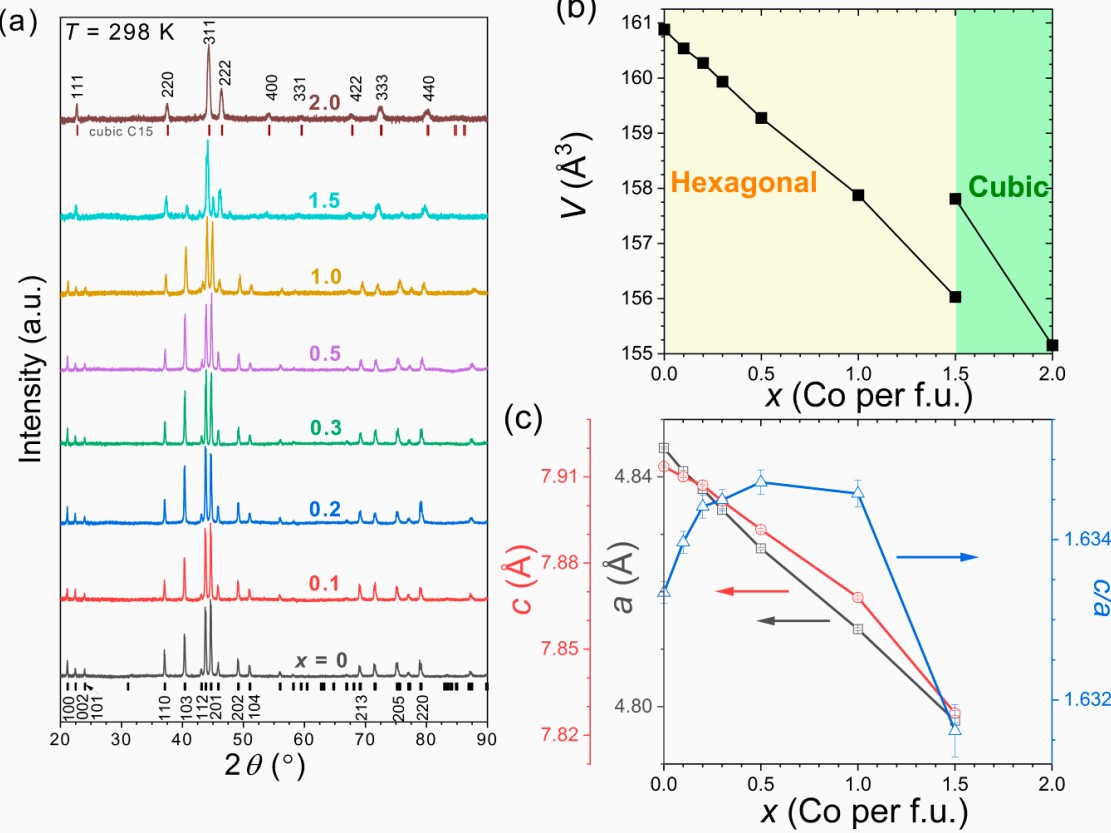

**Figure 3.** Crystal structure of $Sc_{0.28}Ti_{0.72}Fe_{2-x}Co_x$ alloys at room temperature as a function of Co content ($x$): (**a**) powder diffraction patterns measured at room temperature—the ticks for the bottom and upper patterns mark out the reflections for hexagonal C14 and cubic C15 structures, respectively; (**b**) cell volume (half-cell volume for the cubic phase); (**c**) lattice parameters of the hexagonal phase.

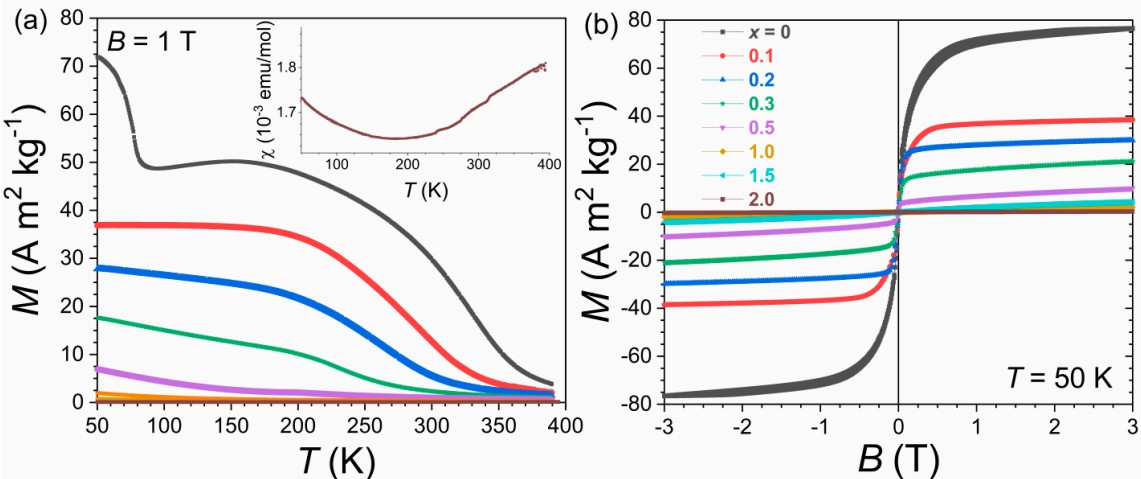

**Figure 4.** Magnetic properties of $Sc_{0.28}Ti_{0.72}Fe_{2-x}Co_x$ alloys as a function of Co content ($x$): (**a**) magnetization versus temperature upon heating in $B = 1$ T, in inset magnetic susceptibility for $x = 2$; (**b**) hysteresis loop at $T = 50$ K.

In $(Sc,Ti)Fe_2$ and $(Sc,Ti)Fe_{1.95}$, a typical signature of the Fe moment fluctuation at $T_{t1}$ was a large cell-volume contraction $\Delta V/V \sim -1.1\%$ upon heating [15–17]. Powder X-ray diffraction experiments were carried out in $Sc_{0.28}Fe_{0.72}Fe_{1.9}Co_{0.1}$ between 100 and 480 K; see Figure 5. The absence of noticeable cell-volume anomaly further confirms the absence of the $T_{t1}$ transition in Co-substituted samples (in the investigated temperature range). The volume coefficient of thermal expansion $\sim +15$ ppm/K is nearly constant from 100 to 330 K, then increases in the paramagnetic phase +25 ppm/K, which is close to former reports on ternary alloys [15,16].

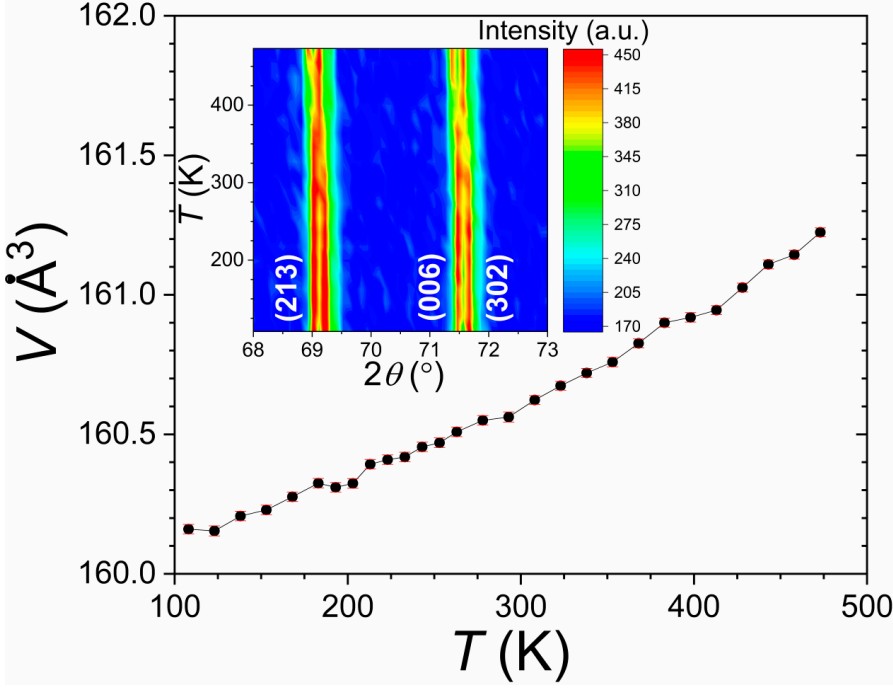

**Figure 5.** Temperature dependence of X-ray diffraction for $Sc_{0.28}Fe_{0.72}Fe_{1.9}Co_{0.1}$.

### 3.3. Mn for Fe Substitution in $Sc_{0.28}Ti_{0.72}Fe_2$

Figure 6 presents the crystal structure of $Sc_{0.28}Ti_{0.72}Fe_{2-y}Mn_y$ alloys determined from powder X-ray diffraction experiments. All alloys crystallize in the C14 crystal structure, so that Mn for Fe

substitutions form a continuous solid solution. Some scatter on the cell-volume vs. composition curve is noticeable, most likely originating from two challenges met during the synthesis: (*i*) Mn evaporation occurs during the arc-melting process, which is only imperfectly compensated for by the addition of extra-manganese; (*ii*) Mn substitutions lead to the formation of a minor cubic impurity phase resembling FeTi alloys (up to ~4 w% in $y = 1$), on which annealing at 1000 °C has only minor effect. Nevertheless, Mn for Fe substitutions clearly show a cell-volume increase (+2.2%) due to an expansion of both *a* and *c* axes, but slightly anisotropic with an increase in the *c/a* ratio (+0.5%).

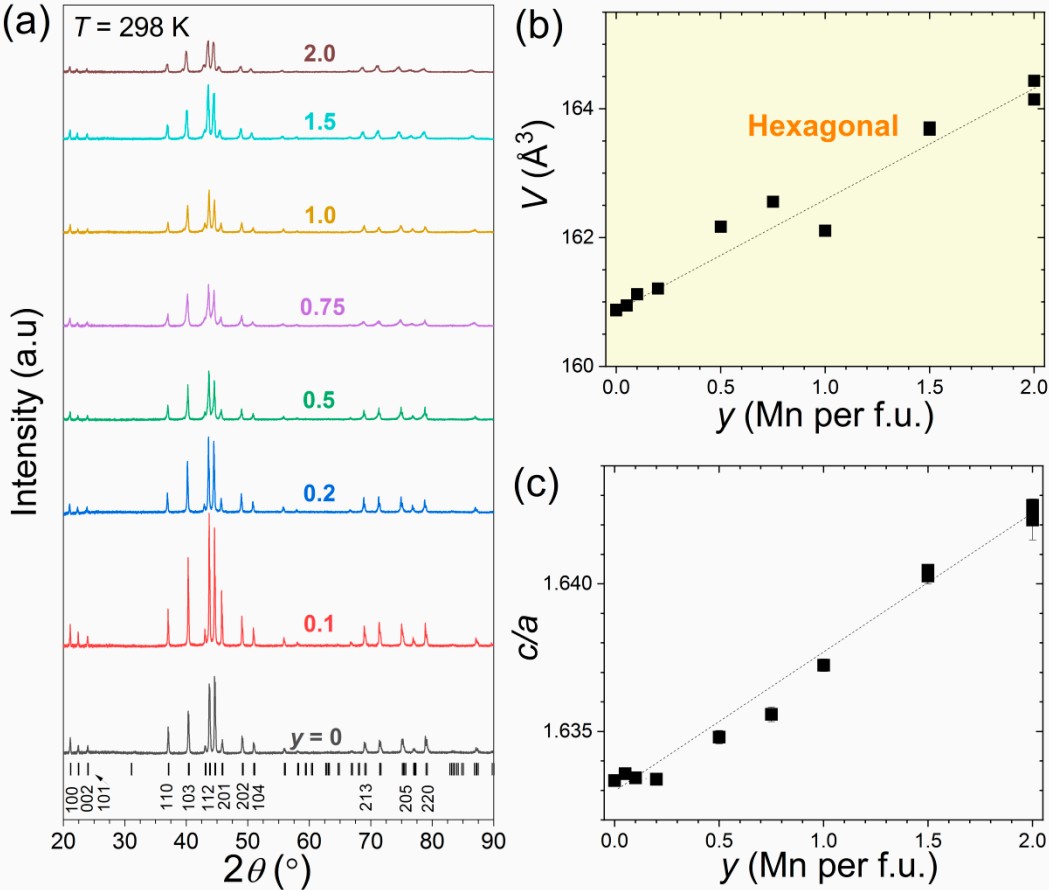

**Figure 6.** XRD patterns and structural parameters at room temperature of $Sc_{0.28}Ti_{0.72}Fe_{2-y}Mn_y$ alloys as a function of Mn content (*y*): (**a**) XRD patterns measured at room temperature, with, at the bottom, ticks marking the reflections of the C14 hexagonal structure; (**b**) unit cell volume; and (**c**) ratio of the cell parameters.

Figure 7 shows the magnetic properties of $Sc_{0.28}Ti_{0.72}Fe_{2-y}Mn_y$ alloys. Mn substitutions lead to a decrease in Curie temperature, in magnetization, and a disappearance of the $T_{t1}$ transition. Mn substitutions also drives the development of the coexisting antiferromagnetic-like phase at $T_f$ toward lower temperatures, so that the magnetization in the finite magnetic field is found to increase around 100 K from $y = 0$ to $y = 0.2$. From $y = 0.5$ and above, the magnetization swiftly decreases; for Mn content larger than $y \geq 1$, a long-range ferromagnetic order is no longer observed. In contrast to $Ti(Fe,Mn)_2$ alloys [40], no clear signature of bulk long-range antiferromagnetic order is found in the present $Sc_{0.28}Ti_{0.72}Fe_{2-y}Mn_y$ samples. The fully substituted sample $Sc_{0.28}Ti_{0.72}Mn_2$ does not exhibit a linear $1/\chi$ Curie–Weiss behavior and is closer to a nearly temperature-independent paramagnet with $\chi_0 = 10^{-3}$ emu·mol$^{-1}$. This is in line with the "non-magnetic" Pauli paramagnetism reported for its $TiMn_2$ and $ScMn_2$ parents [41,42].

Figure 8 summarizes the magnetic data for $Sc_{0.28}Ti_{0.72}Fe_{2-x}Co_x$ and $Sc_{0.28}Ti_{0.72}Fe_{2-y}Mn_y$ alloys. Both Co or Mn substitutions lead to a decrease in Curie temperature and a disappearance of the $T_{t1}$

transition. The magnetization at 50 K and 3 T as a function of Mn or Co substitutions is presented in Figure 8b in terms of average valence electrons per $2a$ and $6h$ sites. Substitutions for Fe lead to a swift reduction in magnetization. The negative chemical pressure due to Mn substitution on Fe does not lead to an increase in $T_{t1}$. Structural parameters such as cell volume are unlikely to be the primary driving force for the control of transition temperature or for the magnetization reduction, as Mn for Fe or Co for Fe substitutions should have resulted in opposite evolutions. An influence of the electron count on moment formation and ordering temperature is anticipated in such an itinerant system. The curve in Figure 8b vaguely resembles the Slater–Pauling curve for binary alloys of $3d$ elements; however, the slopes are much sharper than 1 or −1, preventing the description of the present alloys in a rigid band model. In other respects, the present results differ from that obtained in $Sc_{1-z}Ti_zFe_2$ ternaries. In these, high Fe moments are observed with a titanium content varying from $z = 0.75$ to 0, with a salient $T_{t1}$ transition in the range $0.75 \leq z \leq 0.5$ [17,20,22]. The $d$-electron count of the $A$ atom in $AFe_2$ has a clear influence on the existence of $T_{t1}$ transition and on the transition temperature, with a finite range of possible compositions. By contrast, only 0.1 Mn or Co substitutions for Fe immediately led to the disappearance of the high moment state. The nonsimilar effect of Sc/Ti ratio or substitutions for Fe highlights that other mechanisms are involved in the occurrence of $T_{t1}$ than only the total electron count per f.u.

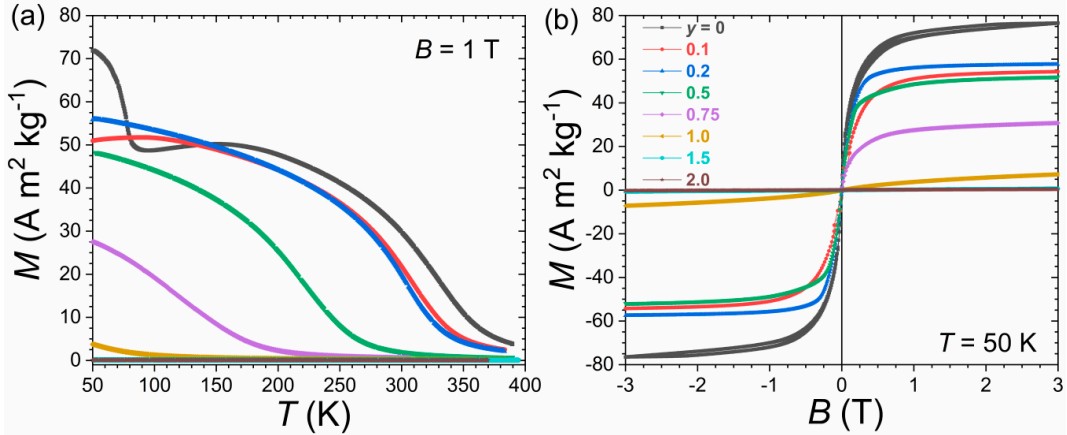

**Figure 7.** Magnetic properties of $Sc_{0.28}Ti_{0.72}Fe_{2-y}Mn_y$ alloys as a function of Mn content ($y$): (**a**) $M(T)$ in $B = 1$ T upon heating, and (**b**) $M(B)$ curves at $T = 50$ K.

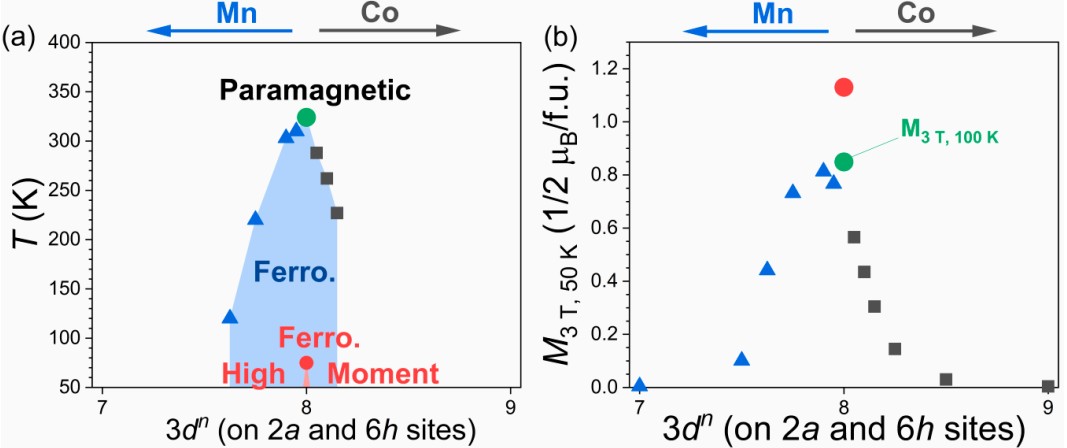

**Figure 8.** (**a**) Transition temperatures for $Sc_{0.28}Ti_{0.72}Fe_{2-y}Mn_y$ and $Sc_{0.28}Ti_{0.72}Fe_{2-x}Co_x$ alloys; (**b**) magnetization at 3 T and 50 K for $Sc_{0.28}Ti_{0.72}Fe_{2-y}Mn_y$ (triangles) and $Sc_{0.28}Ti_{0.72}Fe_{2-x}Co_x$ (squares). For $Sc_{0.28}Ti_{0.72}Fe_2$, the $T_{t1}$ transition temperature and the magnetization above $T_{t1}$ are also indicated (circles).

## 4. Conclusions

The magnetocaloric and magnetoresistance effects of $Sc_{0.28}Ti_{0.72}Fe_2$ Laves phase were investigated. Both effects were found to be substantially larger at the first-order ferro-ferromagnetic transition ($T_{t1}$) than near $T_C$. Though sizable, the magnetocaloric effect remains limited in comparison to other classes of giant magnetocaloric materials. We suggest a strategy to improve these properties by bringing the $T_{t1}$ transition temperature close to $T_C$ and testing its possible realization by Co or Mn for Fe substitutions. Substitutions of Fe by adjacent elements Mn and Co, both of which do not carry a significant magnetic moment of their own in these Laves phases, give rise to a breakdown of the long-range ferromagnetic order, as well as a disappearance of the finite moment magnetism. Substitutions for Fe were found to be dissimilar to the effect of the Sc/Ti ratio, which indicates that other mechanisms than only the total electron count are involved in the occurrence of the ferro-ferromagnetic transition $T_{t1}$, keeping the door open for future independent control of the $T_{t1}$ and $T_C$ transition temperatures.

**Author Contributions:** Investigation, L.S., H.Y., O.T. and F.G.; supervision, F.G. All authors have read and agreed to the published version of the manuscript.

**Funding:** This research was funded by the National Science Foundation of China, grant nos. 51850410514, 51961033, and 11904188; the Inner Mongolia Normal University, grant nos. CXJJS19102, 2018YJRC002, and 2018YJRC003; and supported by the Program for Young Talents of Science and Technology in Universities of Inner Mongolia Autonomous Region NJYT-20-A17.

**Conflicts of Interest:** The authors declare no conflict of interest.

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
