# Peer review of "Magnetocaloric Effect, Magnetoresistance of Sc0.28Ti0.72Fe2, and Phase Diagrams of Sc0.28Ti0.72Fe2−xTx Alloys with T = Mn or Co"

_crystals, doi:10.3390/cryst10050410_

Round 1
Reviewer 1 Report
In this manuscript, the authors study the magnetocaloric and magnetoresistance properties of a Sc-Ti-Fe alloy associated with a first-order magnetostructural transition at low temperature and associated with the second-order ferromagnetic transition at the Curie temperature near room temperature. Magnetocaloric effects, which are found to be small, are calculated using the indirect method via the Maxwell relation, from temperature- and magnetic field-dependent magnetization, whereas magnetoresistance is determined from temperature- and magnetic field-dependent resistivity measurements. Then, the authors want to find a related composition that could bring together the first-order and second-order transitions, with the aim of obtaining enhanced magnetocaloric and magnetoresistance effects. For this purpose they study the dependence of magnetism and structure in quaternary Sc-Ti-Fe-Co and Sc-Ti-Fe-Mn alloys, where Fe is substituted by different concentrations of Mn and Co atoms with respect to the original alloy. The choice is basically made by serendipity because they argue that the large number of parameters playing a role hinder a rational design. It is found that the first-order magnetostructural transition is suppressed in these alloys so that these appear to be unuseful for enhanced magnetocaloric and magnetoresistance properties.
The paper contains results of high quality and discussion that may be of interest, at least, as a valid technical report for material database and for condensed matter physics, so I think that these data deserve publication. However, the way it is presented is a little careless so there are several things that in my opinion should be amended before my personal approval. Below you can find my comments:
- The title of the manuscript claims “phase diagrams of...” but I could not find any phase diagram explicitly plotted. Please, include a temperature-composition phase diagram for each alloy, and a temperature-magnetic field phase diagram for the undoped alloy. With respect to the former, in panels (b) and (d) of Fig. 7 I would indicate somehow that Co concentration increases from the middle to right and Mn concentration increases from the middle to the left. Also, I would indicate Tc for the undoped compound in panel (b) with a blue triangle as done in panel (d).
- I do not see a reason why the structural data concerning the quaternary alloys are presented so different between the two cases (compare figs 3 and 6). Aesthetics is relevant if it helps presenting the data significantly clearer. Also, fig. 7 mixes data concerning the Mn-doped alloy (panels a,c) and concerning the two alloys (b,d), which is confusing, more so when fig. 4 presents the same data as in panels a,c but for the other alloy. I strongly suggest to split fig. 7 into two figures, one similar to fig. 4 and the other for the remaining panels. On the other hand, why there is no temperature-dependent volume from x-ray data concerning the Mn-doped alloy?
- Labels in many figures are too small, see for instance labels for x-axis ticks and y-axis ticks in Fig. 6, numbers in inset in Fig. 5, and in inset of Fig. 4a, etc., alloy composition in Fig. 4b, etc. Why the composition is usually labelled only with “y = ”, but in Fig. 4b the full composition is indicated? Please include (hkl) indices in peaks shown in, at least, one of the x-ray diffraction patterns for each panel. Unit cells in fig. 3b: everything is very badly displayed there.
- Line 46: Specify if “volume contraction” occurs on heating or on cooling.
- Give more details about x-ray experiments: reflection? transmission? capillar width?
- Fig. 1a: I understand that label “delta T” should be replaced by “\Delta B”. Also, indicate in the caption that this corresponds to the magnetostructural transition at Tt1.
- “delta T” is also used in Fig. 1c to determine the temperature step and it might be better to write “Delta T”, with Delta in capital letters. “delta T” is also used in the text (line 99 and 100) to refer to the transition width, but it is not defined explicitly. “delta M” is defined as the magnetic discontinuity at the transition. Such abuse of use of delta for different concepts is confusing, more so when they are not defined explicitly nor rigorously (I cannot see any discontinuity in Magnetization at T_t1).
- For the sake of clarity, indicate in lines 127-129 that x-ray patterns in Fig. 3 are performed “at room temperature”, and place a label “T = 298 K” in fig. 3.
- Line 145: what does T_f mean?
- Line 178: the authors say “Figure 6 presents the evolution of the crystal structure...” but they do not make explicit the evolution as a function of what.
- Fig.7: specify that the numbers in panel (a) refer to the Mn concentration “y”.
Reviewer 2 Report
The paper is rather interesting and deserves to be published in the Magnetocalorics section of Crystals but with obligatory revisions (minor)
- Line 224-225: I failed to find in the text any comparison to other magnetocaloric materials except other Laves phase-type. I recommend adding comparison at least with the highest ever achieved MCE in FOMT materials - FeRh e.g. https://www.sciencedirect.com/science/article/abs/pii/S0304885319334614 it will strengthen the introductory part. Moreover the authors should give an example of other magnetocaloric materials which undergo the first order magnetic transition.
- Fig 1 b and d - I suggest presenting MCE as usual - with the peak up, inspite the negative effect in the studied material. It makes unnecessary unclearance for the reader.
- I would also recommed performing direct MCE measurements on these materials but in case it's experimentally available.
Round 2
Reviewer 1 Report
In their response, the authors have addressed all my concerns appropriately, so that in my opinion the revised version is now clear enough as to deserve publication in Crystals. However, I still have very few comments that I find it necessary to be addressed for a slightly better clarity of the manuscript:
- line 97: Replace “normal” by “conventional”, according to the usual notation. Also, replace “negative” by “with a negative isothermal entropy change” or something similar, as the adiabatic temperature change will be positive (and it is also a magnetocaloric effect).
- line 102: Replace “magnetic” by “magnetocaloric”
- line 170: in the expression “0.32 μB/T.M.”, what does “T.M.” refer to?
- Inset of Fig. 4a: According to the linear behavior in the paramagnetic phase, should the label of the y-axis be the inverse of magnetic susceptibility, instead of just magnetic susceptibility?
- “J. Mag. Mag. Mater.” should be “J. Magn. Magn. Mater.” Other typos are also present.
Author Response
We thank the reviewer for the positive assessment and apologize for the remaining typos.
- line 97: Replace “normal” by “conventional”, according to the usual notation. Also, replace “negative” by “with a negative isothermal entropy change” or something similar, as the adiabatic temperature change will be positive (and it is also a magnetocaloric effect).
Changed as suggested.
- line 102: Replace “magnetic” by “magnetocaloric”
Changed as suggested.
- line 170: in the expression “0.32 μB/T.M.”, what does “T.M.” refer to?
“per transition metal” is replaced by “per Fe or Co atom”
- Inset of Fig. 4a: According to the linear behavior in the paramagnetic phase, should the label of the y-axis be the inverse of magnetic susceptibility, instead of just magnetic susceptibility?
“Susceptibility” is replaced by “magnetic susceptibility”. We confirm we plotted the susceptibility and not the inverse of the susceptibility, as usually done to highlight the presence of a temperature independent term.
- “J. Mag. Mag. Mater.” should be “J. Magn. Magn. Mater.” Other typos are also present.
This misprint is now corrected as well as some other typos in the references.
Reviewer 2 Report
Accept